# Molecular Mechanisms of Dermal Aging and Antiaging Approaches

**DOI:** 10.3390/ijms20092126

**Published:** 2019-04-29

**Authors:** Jung-Won Shin, Soon-Hyo Kwon, Ji-Young Choi, Jung-Im Na, Chang-Hun Huh, Hye-Ryung Choi, Kyung-Chan Park

**Affiliations:** 1Department of Dermatology, Seoul National University Bundang Hospital, Seongnam 13620, Korea; spellbound00@hanmail.net (J.-W.S.); soonhyo17@hanmail.net (S.-H.K.); dhcjy1101@gmail.net (J.-Y.C.); jina1@snu.ac.kr (J.-I.N.); chhuh@snu.ac.kr (C.-H.H.); hyeryung.choi@gmail.com (H.-R.C.); 2Department of Dermatology, Seoul National University College of Medicine, Seoul 03080, Korea

**Keywords:** dermal aging, collagen, fibroblast, elastic fiber, glycosaminglycans, hyaluronic acid, proteoglycans

## Abstract

The dermis is primarily composed of the extracellular matrix (ECM) and fibroblasts. During the aging process, the dermis undergoes significant changes. Collagen, which is a major component of ECM, becomes fragmented and coarsely distributed, and its total amount decreases. This is mainly due to increased activity of matrix metalloproteinases, and impaired transforming growth factor-β signaling induced by reactive oxygen species generated during aging. The reduction in the amount of collagen hinders the mechanical interaction between fibroblasts and the ECM, and consequently leads to the deterioration of fibroblast function and further decrease in the amount of dermal collagen. Other ECM components, including elastic fibers, glycosaminglycans (GAGs), and proteoglycans (PGs), also change during aging, ultimately leading to a reduction in the amount of functional components. Elastic fibers decrease in intrinsically aged skin, but accumulate abnormally in photoaged skin. The changes in the levels of GAGs and PGs are highly diverse, and previous studies have reported conflicting results. A reduction in the levels of functional dermal components results in the emergence of clinical aging features, such as wrinkles and reduced elasticity. Various antiaging approaches, including topicals, energy-based procedures, and dermal fillers, can restore the molecular features of dermal aging with clinical efficacy. This review summarizes the current understanding of skin aging at the molecular level, and associated treatments, to put some of the new antiaging technology that has emerged in this rapidly expanding field into molecular context.

## 1. Introduction

The skin has three layers: The epidermis, dermis, and subcutaneous tissue. With the aging process of the skin, these three components undergo degenerative changes, and changes to the dermis are the most obvious. Skin aging can be classified into two categories: Intrinsic and extrinsic [1]. Intrinsic aging occurs with advancing age and is characterized by fine wrinkles and a thinning epidermis [2,3,4]. In contrast, extrinsic aging is characterized by deep wrinkles, skin laxity, and hyperpigmentation, and is mainly caused by chronic sun exposure [5,6]. Regardless of aging type, wrinkles and reduced elasticity are typical phenomena of skin aging and the result of progressive atrophy of the dermis [7]. One of the main mechanisms of dermal atrophy is thought to be a reduction in the amount of extracellular matrix (ECM), particularly collagen in the dermis [8]. In aged skin, the production of collagen decreases and its degradation increases, which leads to an overall reduction in collagen amount [3,9,10]. Most antiaging approaches target and aim to reverse this process. Structural changes in the collagen and other ECM proteins are also involved in dermal aging. In the present review, we describe the molecular changes in each dermal component during aging and the pathways underlying these changes. We also critically discuss current antiaging approaches for dermal rejuvenation.

## 2. Composition of the Dermis

Unlike the epidermis, which is made up of dense keratinocytes, the dermis is comprised primarily of an acellular component, the ECM. Collagen fibers are a major component of the ECM, accounting for 75% of the dry weight of skin, and provide tensile strength and elasticity. In human skin, type I collagen makes up 80 to 90% of the total collagen, while type III makes up 8 to 12%, and type V makes up <5%. Normally, the collagen bundles increase in size deeper in the dermis.

Elastic fibers are another fibrous element that make up the dermal ECM. Elastic fibers return the skin to its normal configuration after being stretched or deformed. The other components of the ECM are proteoglycans (PGs) and glycosaminoglycans (GAGs), which are amorphous and surround and embed the fibrous and cellular matrix elements in the dermis. Although they only make up 0.2% of the dry weight of the dermis, they absorb water up to 1000 times their volume, and have roles in the regulation of water-binding and compressibility of the dermis.

Fibroblasts are dermal-resident cells, and are differentiated from mesenchymal cells. They are responsible for the synthesis and degradation of fibrous and amorphous ECM proteins. Their function and interaction with the environment are important to understand the molecular mechanism of dermal aging. Other cellular components of the dermis include immune cells like histiocytes, mast cells, and dermal dendrocytes, and endothelial cells and skin appendages, which are beyond the scope of this review.

## 3. Changes in Dermal Components with Aging

### 3.1. Collagen

Quantitative and structural changes in collagen fibers are the major modifications found in aged skin [11,12,13]. In contrast to those in young skin, which has abundant, tightly packed, and well-organized intact collagen fibrils, collagen fibrils in aged skin are fragmented and coarsely distributed [12,14]. Previous studies have shown that increased collagen degradation and reduced collagen biosynthesis are both involved in this aberrant collagen homeostasis, which results in a net collagen deficiency (Figure 1) [11,15,16]. This process leads to clinical changes, such as skin wrinkling and loss of elasticity, which are observed in both naturally and photoaged skin [11,17].

#### 3.1.1. Increased Matrix Metalloproteinase (MMP) Levels

MMPs are a family of ubiquitous endopeptidases that can degrade ECM proteins [18]. MMPs can be categorized into five main subgroups, namely: (1) Collagenases (MMP-1, MMP-8, and MMP-13); (2) gelatinases (MMP-2 and MMP-9); (3) stromelysins (MMP-3, MMP-10, and MMP-11); (4) matrilysins (MMP-7 and MMP-26); and (5) membrane-type (MT) MMPs (MMP-14, MMP-15, and MMP-16) [19]. MMP-1 is the major protease that initiates fragmentation of collagen fibers, which are predominantly type I and III in human skin. After cleavage by MMP-1, collagen can be further degraded by MMP-3 and MMP-9 [20,21]. In the skin, the major source of MMPs are epidermal keratinocytes and dermal fibroblasts, although MMPs can also be produced by endothelial cells and immunocytes [22,23]. Physiologically, MMPs are regulated by the specific endogenous tissue inhibitors of metalloproteinases (TIMPs), which compose a family of four protease inhibitors: TIMP-1, TIMP-2, TIMP-3, and TIMP-4 [24].

Previous studies have demonstrated that levels of MMP-1, MMP-2, MMP-3, MMP-9, MMP-10, MMP-11, MMP-13, MMP-17, MMP-26, and MMP-27 are elevated in aged human skin [25,26,27,28]. MMPs and TIMPs are often regulated in coordination to control excess MMP activity. However, elevation of MMP levels in aged skin is not accompanied by a corresponding increase in the levels of endogenous MMP inhibitors [25,26]. The amount of TIMP-1 in photoaged and intrinsically aged skin may even be reduced [29]. This imbalance accelerates progressive collagen fragmentation in the dermis, and accelerates skin aging.

Reactive oxygen species (ROS) are a major driving force behind the increase in MMP levels in aged skin [12,30,31]. ROS are generated in the skin from both extrinsic and intrinsic sources, such as ultraviolet irradiation and metabolically generated pro-oxidants. ROS activate the mitogen-activated protein kinase (MAPK) family, comprised of extracellular signal-regulated kinase (ERK), p38, and c-Jun NH2- terminal kinase (JNK). This activation induces the transcription factor, activator protein 1 (AP-1), which plays an essential role in the transcriptional regulation of MMP-1, MMP-3, MMP-9, and MMP-12 (Figure 1) [32,33,34,35]. Nuclear factor-κB (NF-κB) is another transcription factor that is activated by ROS [36]. Critically, NF-κB mediates the responses to UV irradiation and photoaging. NF-kB activity is responsible for the upregulation of MMPs such as MMP-1 and MMP-3 in dermal fibroblasts [19,34,37,38]. Generally, oxidative damage is more obvious in photoaged skin, and this may explain more prominent associated aging features like deep wrinkles. While the primary source of MMPs in intrinsic aging are dermal fibroblasts, MMPs in photoaging are also produced by epidermal keratinocytes [19,26].

#### 3.1.2. Impaired Transforming Growth Factor-β Signaling during Aging

Transforming growth factor-β (TGF-β) is a major regulator of ECM biosynthesis [39]. In human dermal fibroblasts, TGF-β controls collagen homeostasis by regulating both collagen production and degradation via the Smad pathway [16]. Initially, TGF-β binds to a TGF-β type II receptor (TβRII), which recruits and phosphorylates a TGF-β type I receptor (TβRI). This phosphorylation of TβRI leads to activation of the transcription factors Smad2 and Smad3. Activated Smad2 or Smad3 combines with Smad4 to form heteromeric Smad complexes. These activated Smad complexes translocate into the nucleus and interact with Smad-binding elements (SBE) in the promoter regions of TGF-β target genes [40,41,42]. ECM genes including collagens, fibronectin, decorin, and versican are thus directly upregulated by TGF-β/Smad signaling. In contrast, MMPs are downregulated and TIMPs are upregulated by the Smad signaling network. This suggests that the TGF-β/Smad signaling pathway is critical for maintaining the structural and mechanical integrity of dermal connective tissue by enhancing ECM production and inhibiting ECM degradation.

In aged skin, AP-1 induced by ROS inhibits the TGF-β signaling pathway in dermal fibroblasts (Figure 1). Several previous studies have demonstrated that specific down regulation of TβRII and SMAD3 expression may be involved in decreased TGF-β signaling [43]. Impaired TGF-β signaling leads to decreased synthesis of neocollagen, and results in a reduction in net collagen amount in the dermis.

#### 3.1.3. Interaction between Fibroblasts and the ECM

In young skin, fibroblasts adhere to the surrounding intact ECM, which is mainly comprised of type I collagen [15,44]. This adherence allows fibroblasts to exert mechanical forces on the surrounding ECM, and to spread and maintain a normal elongated shape. In aged skin, fibroblast attachment is impaired due to progressive ECM degradation, resulting in fibroblast size reduction, decreased elongation, and collapsed morphology (Figure 1) [3,15,25,45,46]. Reduced size is a key feature of senescent fibroblasts, and is correlated with decreased production of ECM components [25]. The reduction of dermal fibroblast spreading and cell size can also increase mitochondrial ROS generation (Figure 1) [47]. Reduced fibroblast size and mechanical forces specifically downregulate TβRII, and this downregulation largely mediates the reduction in TGF-β-regulated ECM production [45]. Furthermore, the reduction in fibroblast size regulates ECM degradation through elevation of MMP levels. Recently, Quin et al. proposed a mechanism by which age-related reduction in fibroblast size activates AP-1, which in turn induces the production of multiple MMPs as observed in aged human skin. Elevated ROS levels in fibroblasts with aged features may mediate this pathway [26].

#### 3.1.4. Summary

ROS generated in the aging process increase MMP expression and inhibit TGF-β signaling, which leads to collagen fragmentation and decreased collagen biosynthesis (Table 1). This hinders the mechanical interaction between fibroblasts and the ECM, and consequently leads to a reduction in the size of dermal fibroblasts. The aged fibroblasts produce more ROS, which further increase the expression of MMPs and inhibit TGF-β signaling, creating a positive feedback loop that accelerates dermal aging (Figure 1). Interestingly, fibroblasts in aged skin recovered their appearance and function when in contact with intact ECM [48] implying that the ECM microenvironment is a major factor in the aging of fibroblasts.

### 3.2. Changes to other ECM Components

#### 3.2.1. Elastic Fiber Remodeling

Elastic fibers play an important role in skin compliance (ability to be readily deformable) and resilience (ability to recoil), which together are often referred to as skin elasticity [49,50,51]. In the dermis, they are produced mainly by fibroblasts [52]. Elastic fibers are formed from soluble tropoelastin molecules, an elastin precursor, via cross-linking by lysyl oxidase (LOX) [53,54]. They consist of elastic microfibrils (such as numerous proteins such as fibrillin, fibulin, and microfibrillar-associated glycoproteins (MAGPs], and latent TGF-β -binding proteins (LTBPs)) and amorphous elastin [55,56,57]. In young skin, the elastic fibers adopt a characteristic highly ordered architecture, with perpendicularly oriented fibrillin-rich microfibrils at the papillary dermis and large-diameter elastic fibers in the reticular dermis, comprised primarily of elastin [58].

During the aging process, the elastic fiber system undergoes structural changes. In intrinsic skin aging, fibrillin-rich microfibrils in the papillary dermis are selectively degraded. Fibulin-5 plays an important role in the association of tropoelastin, a precursor of elastin, with microfibrils, to form elastic fibers [59]. There is evidence that fibulin-5 is related to elastic fiber remodeling in intrinsic skin aging [49,60]. In young photoprotected skin, fibulin-5 is localized to the elastic fiber throughout the dermis; however, in aged skin, this pattern of fibulin-5 is completely abolished. In contrast to intrinsic aging, photoaging is characterized by accumulation of disorganized elastic fibers throughout the dermis in a process termed “solar elastosis” [51]. The main mechanism responsible for increased elastic fiber degradation in photoaging is the activation of MMPs [11]. MMPs-2, -3, -9, -12, and -13 are capable of catabolizing elastic fibers [61,62]. MMP-12, also known as human macrophage metalloelastase, is the most active protease in elastin degradation (Figure 1) [63]. Chung et al. demonstrated that the induction of MMP-12 gene and protein expression caused by UV radiation contributed to the development of solar elastosis in human skin [64]. In addition, Imokawa et al. also showed that repetitive UV radiation at suberythemal doses induced upregulated activity of skin fibroblast-derived elastase and impairment of elastic fiber configuration, and the subsequent loss of skin elasticity [65]. In photoaged skin, replenishment of the degraded matrix also decreases. Weiss showed a reduction in elastin production with age, or as a result of photodamage. Cenizo et al. reported that the expression of LOX and LOX-like enzymes decreased with aging, resulting in a reduction in the assembly of new elastic fibers [66]. Solar elastosis is characterized by an increase of elastic material in the dermis, which seems to be a paradoxical situation. However, the coarse and disoriented fibers observed in solar elastosis are essentially nonfunctional [52]. In summary, with aging, the number of functional elastic fibers is reduced, and this is related with functional changes in aged skin such as loss of elasticity and wrinkle formation [51].

#### 3.2.2. Changes in Glycosaminoglycans

Glycosaminoglycans (GAGs) are large linear polysaccharides, and are a major component of the ECM. There are six types of GAGs, including chondroitin sulfate (CS), dermatan sulfate (DS), keratan sulfate (KS), heparan sulfate (HS), heparin (HP), and hyaluronic acid (HA) [67,68,69]. Except HA, GAGs contain sulfate substituents at different positions on the chain, and are heavily glycosylated on their related proteoglycan (PG) core proteins [69]. HA is not sulfated and does not attach to proteins to form proteoglycans. Instead, it binds to proteins containing HA-binding domain [69]. Because GAG chains contain numerous negatively charged carboxyl and sulfate groups, they may have important roles in maintaining the water content in tissue. Crosslinking of HA with matrix proteins such as the collagen network results in the formation of supermolecular structures and increases tissue stiffness [70].

Dermal HA is mainly produced by fibroblasts, and it is abundant in the papillary dermis [68]. It crosslinks with other ECM proteins, like collagen, resulting in increased tissue stiffness, and plays space filling and shock-absorbing roles [70]. In intrinsically aged skin, HA binding proteins (HABPs) are reduced compared with in young skin, while the level of HA itself is not significantly different between young and old skin [71,72]. HABPs are known to trigger several intracellular signaling pathways regulating proliferation, migration, and differentiation [73]. In contrast, dermal HA content in photoaged skin is significantly increased, particularly in regions of solar elastosis [73]. Although UV irradiation induces HA synthase (HAS) [68,74], HAS mRNA levels in aged sun-exposed skin were significantly reduced compared to those in sun-protected skin, indicating the possibility of unknown regulatory mechanisms at work [69,75]. Like solar elastosis, increased HA in photoaged skin might be the result of abnormal accumulation of nonfunctional proteins.

In a previous study, total sulfated GAGs were found to be decreased during intrinsic aging [72]. However, these changes may be specific to each type of sulfated GAGs: HS and CS decrease, while KS and DS increase [69]. In photoaged skin, total sulfated GAGs increased, similarly to HA [72]. CS staining increased in the solar elastosis region [73]. Further investigations are necessary to reveal the changes to specific sulfated GAGs, and the underlying mechanisms.

#### 3.2.3. Changes to Proteoglycans

PGs are a family of GAG conjugated proteins, and are essential for maintaining the mechanical strength of the skin [76]. Previous work has shown that decorin, biglycan, and versican are the most abundant proteoglycans in human skin. Decorin and biglycan are core proteins of DS, while versican is a core protein of CS. Several previous studies have investigated changes to these major PGs in intrinsically aged skin, with results varying depending on the location, gender of the subjects, and detection methods [69]. In photoaged skin, versican seems to be increased, while biglycan does not change, and decorin is absent [69,77]. There are complex mechanisms regulating these changes in PGs with aging, and further investigations are warranted.

#### 3.2.4. Summary

In intrinsic aging, the amount of elastic fibers is reduced; however, in extrinsic aging, nonfunctional, abnormal elastic fibers accumulate in the papillary dermis (Table 1). Changes to the amount of GAGs and PGs vary in intrinsically aged skin, but these components generally increase with aging, with the exceptions of decorin and biglycan in photoaged skin (Table 1). Any increase observed in photoaged skin seems to be the result of abnormal accumulation or through compensational recovery responses for cumulative damage to the ECM [69].

## 4. Antiaging Approaches

Antiaging strategies have been developed with the stated aim of achieving a young and healthy skin. Here, currently widely used antiaging approaches with demonstrated dermal rejuvenation and the associated molecular mechanisms will be reviewed briefly (Table 2).

### 4.1. Topicals

#### 4.1.1. Topical Retinoids

Topical retinoids are still considered the gold standard for clinically effective topical antiaging products [110]. Retinoids are a family of compounds composed of vitamin A, its derivatives, and synthetic molecules acting through the same pathway [82]. Besides all-trans retinoic acid (RA), the major bioactive form, the term “retinoids” includes retinaldehyde, retinol, and various retinyl esters. They act through retinoic acid receptors (RARs) and retinoid X receptors (RXRs), increasing the amount of type I procollagens and decreasing the amount of MMPs (Table 2) [78,80]. RA increases the amount of type I, III, and VII collagens in the dermis, and can reorganize the dermal collagen into new woven bundles [79]. In addition, RA stimulates normalization of the elastic tissue organization and GAG deposition in the dermis [81]. In clinical trials, topical retinoids are clinically effective for the treatment of dermal aging, including wrinkles, roughness, and laxity (Table 2) [83,111,112,113,114]. However, they can be irritating to a significant percentage of the population, and further development of ways to reduce their irritation potential is needed [115]. Retinol, which metabolizes to retinaldehyde and retinoic acid, has been found to be less irritating than retinoic acid and is widely used as an ingredient for antiaging cosmeceuticals [116,117].

#### 4.1.2. Other Cosmeceuticals

Oxidative stress initiated by ROS generation is an important factor modulating dermal alteration in the aging process. Topical antioxidants enhance resistance to oxidative stress and prevent dermal damage and consequently reduce the rate of skin aging [84]. Ascorbic acid (vitamin C) has been used widely as an antiaging and hyperpigmentation topical agent for several decades. Ascorbic acid eliminates most ROS due to the oxidation of ascorbate to monodehydroascorbate, and then to dehydroascorbate, and can maintain the normal physiological state of human skin [84]. In the skin, ascorbic acid is a cofactor required for the synthesis of procollagen and elastin [85]. Ex vivo and in vivo studies suggest that ascorbic acid induces collagen synthesis in human skin fibroblasts, and increases dermal thickness (Table 2) [86,87]. Topical formulations containing ascorbic acid have clinical efficacy in antiaging treatments (Table 2) [71,88,118]. However, poor skin penetration and chemical instability reduce the clinical efficacy of ascorbic acid, and require additional research.

Topical α-hydroxy acids also showed clinical efficacy in the treatment of photoaged skin [89,119]. Glycolic or lactic acid at 5% to 25% concentration stimulates GAGs and collagen production in the dermis and improves the histologic quality of elastic fibers (Table 2) [89,115].

Peptides are chemical compounds composed of short chains of amino acids. Due to their small size, they can penetrate the upper layer of the skin [93]. They act as dispatchers that trigger specific functions, such as regulating fibroblasts and controlling the production of ECM components (Table 2) [90,91]. For these reasons, peptides have recently attracted great interest in the cosmeceutical industry and showed clinical efficacy in several trials (Table 2).

### 4.2. Energy Based Dermal Rejuvenation

#### 4.2.1. Fractional Lasers

The basic concept for dermal rejuvenation by fractional lasers is the laser-induced wound healing process. Immune cell influx and increased inflammatory mediators after laser wounding induce the expression of MMPs that degrade old abnormal ECM and stimulate the production of new ECM [94]. Traditionally, fully ablative laser resurfacing remains a gold standard procedure for treatment of photodamaged skin [120]. However, due to concerns regarding down-time and the potential for significant complications, such as scarring and dyspigmentation, alternative less-invasive approaches have been developed [121].

The concept of fractional photothermolysis was recently introduced, and has been widely used as a laser antiaging treatment [94,122]. Fractional laser resurfacing involves microscopically small beams of energy applied to the skin, and treats a small “fraction” of the skin each session, leaving areas between each exposed area. In this way, deeply penetrating columns of laser energy heat the dermis and stimulate matrix remodeling, while allowing for a more favorable safety profile and rapid healing [94]. Carbon dioxide fractional laser therapy is a recently introduced fractional ablative resurfacing modality, with demonstrated clinical improvements to photoaged skin (Table 2) [96,123]. After fractional CO_2_ laser treatment, the expression of MMP-1, -3, and -9 significantly increased temporally, followed by induction of type I and III protocollagen [94,95]. Erbium:YAG, yttrium-scandium-gallium garnet (YSGG) and Er:YSGG (2790-nm) systems are alternative ablative fractional lasers with significant antiaging results (Table 2) [97]. The first nonablative fractional laser was an erbium: glass fractional laser (Table 2) [122]. Although this device primarily targets dermal water, which causes collagen heating and dermal remodeling like ablative lasers, epidermal injury and tissue vaporization do not occur, resulting in an improved safety profile [124,125].

#### 4.2.2. Non-Laser-Based Approaches

Radiofrequency (RF) produces heat when the electrical resistance of tissues converts the electric current to thermal energy. Monopolar RF therapy delivers uniform heat at a controlled depth to the dermal layers, causing direct collagen contraction and immediate skin tightening [99,100]. Subsequent remodeling and reorientation of collagen bundles, and the formation of new collagen, is achieved over months of treatment, which lead to an improvement in skin laxity [101]. Monopolar RF induces an increase in the mean amount of collagen types I and III, and improved the quality of elastic fibers and solar elastosis (Table 2) [102]. It also has demonstrated clinical improvement of skin tightening, texture, and wrinkles [13,103,104,126,127]. The rejuvenation effect of bipolar RF has also been demonstrated, with bipolar RF treatment inducing collagen synthesis, and has clinical efficacy for treating dermal aging (Table 2) [128].

High-intensity focused ultrasound (HIFU) is an evolving technology that has changed rapidly over the past five years [124]. HIFU creates precision microwounds in the dermis without affecting the epidermis using varied high-frequency ultrasound waves [105]. HIFU induced higher levels of neocollagenesis and neoelastogenesis in the deep reticular dermis compared with that induced by monopolar RF [106]. HIFU is effective in patients with mild to moderate skin laxity, with proven improvement to skin subjectively and objectively (Table 2) [107,108,129].

### 4.3. Filler as an ECM Microenvironment Modulator

Dermal fillers are increasingly used for the rejuvenation of the face or hands. They fill rhytides and folds, and replace soft tissue volume that is lost during chronological skin ageing [130], making the skin appear younger. However, recent evidence has suggested that fillers enhance the structural support of the ECM and restore the capacity of fibroblasts in aged human skin (Table 2) [48,109]. Changes to the mechanical interaction between fibroblasts and the ECM cause size reduction in fibroblasts and consequently result in functional abnormalities. Dermal fillers can restore the contractile properties of aged fibroblasts to the same level as that of young normal fibroblasts, and may recover the elongation of fibroblasts. Furthermore, dermal HA fillers induce type I collagen synthesis through the activation of the TGF-β signaling pathway in fibroblasts, as well as increase fibroblast proliferation [48]. Further investigations to elucidate the clinical implications of this phenomenon are needed.

## 5. Conclusions

Dermal components, including fibroblasts, collagen, elastic fibers, GAGs, and PGs undergo significant changes during the intrinsic and extrinsic aging processes. A significant reduction in the amount of collagen induces dermal fibroblasts to acquire age-related features like decreased size and loss of elongation, which consequently result in the deterioration of normal functions like the induction of collagen production and prevention of collagen degradation. Elastic fibers, GAGs, and PGs also change with aging, leading to decreased amount of functional components and clinical aging features like wrinkles and reduced elasticity. Various antiaging approaches, including topicals, energy-based procedures, and dermal fillers can restore the molecular features of dermal aging. Understanding the molecular mechanisms of skin aging helps us to identify novel targets for antiaging treatments.

## Figures and Tables

**Figure 1 ijms-20-02126-f001:**
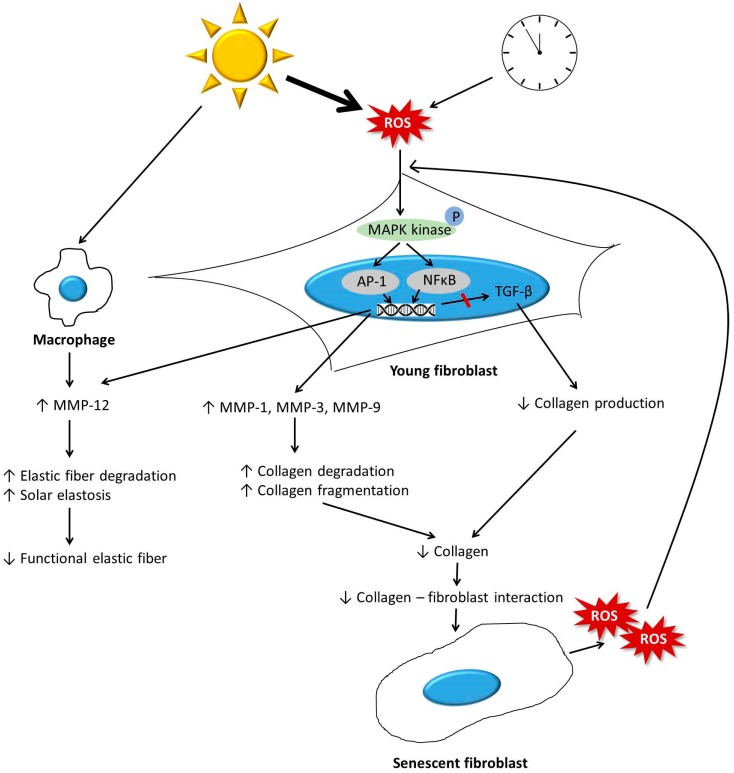
Schematic illustration showing the changes in fibroblasts, collagen, and elastic fibers in the dermal aging process. Reactive oxygen species (ROS) generated in the aging process activate mitogen-activated protein kinases (MAPKs) and induce transcription factors, including activator protein 1 (AP-1) and nuclear factor-κB (NF-κB). This activation increases matrix metalloproteinase (MMP) expression and inhibits transforming growth factor-β (TGF-β) signaling, which leads to collagen fragmentation and decreased collagen biosynthesis. This hinders the mechanical interaction between fibroblasts and the extracellular matrix (ECM), and consequently reduces the size of dermal fibroblasts. Aged fibroblasts produce a greater amount of ROS that further increases the expression of MMPs and inhibits TGF-β signaling, creating a positive feedback loop that accelerates dermal aging. MMP-12 secreted from fibroblasts and macrophages plays a crucial role in the development of solar elastosis and in the reduction of functional elastic fibers.

**Table 1 ijms-20-02126-t001:** Changes in dermal extracellular matrix components in the aging process.

Photoaging	Components	Intrinsic Aging
Decreased and fragmented	Collagen	Decreased and fragmented
Abnormally accumulated (SE)	Elastic fiber	Decreased
Increased in SE region	Hyaluronic acid	Not changed
Increased	Total sulfated GAGs	Decreased
Increased in SE region	Versican	Not changed?
Not changed	Biglycan	Decreased
Decreased in SE region	Decorin	Not changed?

GAG = glycosaminoglycan, SE = solar elastosis.

**Table 2 ijms-20-02126-t002:** Mechanisms of action and clinical efficacies of representative antiaging approaches.

Modalities	Mechanisms of Action	Clinical Efficacies
**Topicals**		
Retinoid acid (RA)	Acts through RARs and RXRs [78] Increases type I, III, and VII collagens [79] Decreases MMPs [78,80] Reorganizes elastic fiber [81] Normalizes GAG deposition [82]	Application of 0.05% RA for 6 months improved fine and coarse wrinkles, roughness, and skin laxity [83]. Application of 0.025% RA for 3 months improved rough and fine wrinkles, skin firmness, and roughness (Ho ET)
Ascorbic acid	Reduces ROS [84] Acts as a cofactor in the biosynthesis of procollagen and elastin [85] Induces collagen synthesis in human skin fibroblasts and increase dermal thickness [86,87]	Application of 5% ascorbic acid for 6 months led to a clinical improvement of the photodamaged skin [88].
Glycolic acid	Stimulates the production of GAGs and collagen in the dermis [89] Improves histologic quality of elastic fibers [89]	Application of 25% glycolic acid for 6 months increased skin thickness [89].
Peptides	Regulate fibroblasts and control the production of ECM [90,91].	Application of Pal-KTTKS for 3 months reduced wrinkles [92]. Application of copper–GHK reduced the depth and length of wrinkles and made skin smoother [93].
**Energy-based devices**		
Fractional lasers (FL)	Heat the dermis and stimulate matrix remodeling by deeply penetrating columns of laser energy [94] Induce biosynthesis of type I and III protocollagens [94,95]	Two or three treatments with CO_2_ fractional laser improved skin texture, laxity, and overall cosmetic outcome [96]. Two treatments with 2790 nm Er:YSGG laser improved wrinkle and skin texture [97]. Three treatments of fractional 1550 nm erbium-doped fiber laser improved wrinkles [98].
Ablative FL		
Nonablative FL		
Radiofrequency (RF)	Causes direct collagen contraction and immediate skin tightening [99,100] Reorganizes collagen bundles [101] Induces increase in types I and III collagens [102] Improves the quality of elastic fibers and solar elastosis [102]	Three treatments with fractional bipolar RF improved wrinkles and skin texture [103]. Six treatments with monopolar RF improved laxity, texture, and wrinkles [104].
High-intensity focused ultrasound (HIFU)	Creates precision microwounds in the dermis [105] Induces the higher level of neocollagenesis and neoelastogenesis in the deep reticular dermis [106]	Single treatment with HIFU improved skin laxity of lower face and neck [107]. Single treatment with HIFU improved skin laxity of face and upper neck [108].
**Fillers**		
	Restore the contractile properties and elongation of aged fibroblasts [48,109] Induce type I collagen synthesis [48]	Further investigation is needed.

ECM = extracellular matrix, ER:YSGG = erbium:yttrium scandium gallium garnet, GAG = glycosaminoglycan, GHK = glycyl-L-histidyl-L-lysyl, MMP = matrix metalloproteinase, pal-KTTKS = palmitoyl pentapeptide palmitoyl-lysine-threonine-threonine-lysine-serine, RAR = retinoic acid receptor, RXR = retinoid X receptors.

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
