# Peer review of "Molecular Mechanisms of Dermal Aging and Antiaging Approaches"

_ijms, 2019, doi:10.3390/ijms20092126_

Reviewer 1 Report

This is a surprisingly readable review of the basic biochemistry and physiology of skin ageing – surprising because many of the dermal changes in aged skin have been well documented decades ago, as is well-illustrated by some references cited here from as long ago as the 1980s.  The topic could have been repetitive and uninspiring. However, the authors have substantially updated the literature with the more recent molecular information that brings the review up to today’s understanding of the subject.  The review will therefore be a very useful contemporary summary of the current knowledge to many readers.

I have only minor comments, mainly typos:

Line 147:  ?numbering for this summary

Line 179: metalloelastase

Line 210:  full stop should be comma

Line 271: ??’chemical’

Author Response

This is a surprisingly readable review of the basic biochemistry and physiology of skin ageing – surprising because many of the dermal changes in aged skin have been well documented decades ago, as is well-illustrated by some references cited here from as long ago as the 1980s.  The topic could have been repetitive and uninspiring. However, the authors have substantially updated the literature with the more recent molecular information that brings the review up to today’s understanding of the subject.  The review will therefore be a very useful contemporary summary of the current knowledge to many readers.

response: Thank you very much for your valuable comments and kind corrections. The manuscript was revised according to your suggestions.

 I have only minor comments, mainly typos:

 Line 147:  ?numbering for this summary

response: The numbering of subtitle ‘Summary’ was changed from 3.1.3 to 3.1.4

Line 179: metalloelastase

response: It was corrected

Line 210:  full stop should be comma

response: The period was corrected to comma.

Line 271: ??’chemical’

response: “chemical” was changed to “chemical instability”

Reviewer 2 Report

Shin et al sumitted a well-written summary of molecular changes of skin aging, focusing on the processes happening in the dermis. I do not have any suggestions for providing more details to the topic. I have a few minor notes:

Please remove the underlining of "elastosis" from Figure 1.

Authors have 3.1.3 paragraph twice (row 133 and row 147)

3. Table 2 should be more organized. I do not see that the ticks would be necessary in front of each statement.

Application OF copper-GHK

At the end of the Conclusion I suggest to add that understanding the molecular mechanisms of skin aging helps us to identify novel targets for anti-aging treatments.

Author Response

Shin et al sumitted a well-written summary of molecular changes of skin aging, focusing on the processes happening in the dermis. I do not have any suggestions for providing more details to the topic. I have a few minor notes:

response: Thank you very much for your valuable comments and kind corrections. The manuscript was revised according to your suggestions now.

 Please remove the underlining of "elastosis" from Figure 1.

response:It was removed.

Authors have 3.1.3 paragraph twice (row 133 and row 147)

response:The latter “3.1.3” was corrected to “3.1.4”

3. Table 2 should be more organized. I do not see that the ticks would be necessary in front of each statement.

response:Thank you for your comment. Table 2 was a bit lengthened for comprehensive description. The ticks in the table were removed.

Application OF copper-GHK

response: “Of” was inserted.

 At the end of the Conclusion I suggest to add that understanding the molecular mechanisms of skin aging helps us to identify novel targets for anti-aging treatments.

response: Thank you for your nice suggestion. The contents were added in the Conclusion now.